# Sinoagetopanorpidae fam. nov., a New Family of Scorpionflies (Insecta, Mecoptera) from the Guadalupian of South China

**DOI:** 10.3390/insects14010096

**Published:** 2023-01-16

**Authors:** Xinneng Lian, Chenyang Cai, Diying Huang

**Affiliations:** State Key Laboratory of Palaeobiology and Stratigraphy, Center for Excellence in Life and Paleoenvironment, Nanjing Institute of Geology and Palaeontology, Chinese Academy of Sciences, Nanjing 210008, China

**Keywords:** diversity, Capitanian, Yangtze Platform, Permochoristidae

## Abstract

**Simple Summary:**

Mecopterans have been sparsely reported from the Permian of China, despite their great biodiversity in the fossil record in the world. Herein, we describe and illustrate three genera (two new genera) and eleven species (ten new species) belonging to a new family Sinoagetopanorpidae fam. nov. from the upper Guadalupian Yinping Formation of Anhui Province, China: *Sinoagetopanorpa permiana* Lin, Nel and Huang, 2010, *S. nigra* sp. nov., *S. rotunda* sp. nov., *S. lini* sp. nov., *S. minuta* sp. nov., *S. elegans* sp. nov., *S. grimaldii* sp. nov., *S. magna* sp. nov., *Raragetopanorpa zhangi* gen. et sp. nov., *Permoagetopanorpa yinpingensis* gen. et sp. nov. and *P. incompleta* sp. nov. Our new discovery indicates a high diversity of mecopterans in the Permian of China, and Signoagetopanorpidae might have evolved independently on the Yangtze Platform.

**Abstract:**

Mecoptera was in great abundance in the Permian, but little is known from China. A new family, Sinoagetopanorpidae fam. nov., is described and illustrated from the upper Guadalupian Yinping Formation at Yinping Mountain, Chaohu City, Anhui Province, China. *Sinoagetopaorpa permiana* Lin, Nel and Huang, 2010 was previously attributed to Permochoristidae and now is revised as the type species of Sinoagetopanorpidae fam. nov. Three genera (two new genera) and ten new species of this new family are described and illustrated: *Sinoagetopanorpa permiana* Lin, Nel and Huang, 2010, *S. nigra* sp. nov., *S. rotunda* sp. nov., *S. lini* sp. nov., *S. minuta* sp. nov., *S. elegans* sp. nov., *S. grimaldii* sp. nov., *S. magna* sp. nov., *Raragetopanorpa zhangi* gen. et sp. nov., *Permoagetopanorpa yinpingensis* gen. et sp. nov. and *P. incompleta* sp. nov. Some isolated hind wings are described and illustrated, although it is difficult to assign them to any particular species. As a dominant mecopteran lineage in the Yinping Formation, Sinoagetopanorpidae represents an endemic group that might have independently evolved on the Yangtze Platform.

## 1. Introduction

The common name “scorpionflies” of the insect order Mecoptera is derived from the fact that some males bear unturned and bulbous genitalia that resemble the stingers of scorpions. Holometabolous insects have the richest biodiversity among all insect clades and can be dated back to the Pennsylvanian [1]. Mecoptera is one of the most ancient and morphologically generalized holometabolous insect orders, and they represent an important holometabolous group in the Permian. Mecoptera can be traced back to the beginning of the Permian [2] and thrived from the Permian to Cretaceous [3,4,5,6,7,8,9,10,11,12,13,14,15,16,17,18,19,20]. However, many mecopteran lineages went extinct during the Cretaceous, and a modern-looking mecopteran assemblage emerged in the Cenozoic. Extant mecopterans encompass ca. 700 species and 40 genera assigned to 9 families [21,22]. Bittacidae and Panorpidae are overwhelmingly diverse among mecopterans in the extant fauna. Mecoptera is more prosperous in the fossil record than it is in the Recent, with more than 700 species and 210 genera in approximately 40 families recorded to date [22,23,24].

Mecopterans were abundant and since the Cisuralian; more than 200 species and 40 genera of Permian mecopterans have been described from Africa, the Americas, Asia, Australia, Europe and India [5,6,25,26,27,28,29,30,31]. Rare specimens of Permian mecopterans in China were described from the Yinping Formation [31,32,33], represented by three reported species: *Sinoagetopanorpa permiana* Lin, Nel and Huang, 2010, *Permica chaohuensis* Lian, Cai and Huang, 2022 and *Chaohuchorista liaoi* Lian, Cai and Huang, 2022. *Sinoagetopanorpa permiana* is peculiar in its broad wing and costal area, and the apically curved R_1_. Lin et al. [31] assigned *S. permiana* to Permochoristidae Tillyard, 1917 originally. However, the holotype of *S. permiana* is slightly deformed, and the line drawings were probably not precise in the original paper. During the past 12 years, we have collected 56 specimens closely related to *S. permiana* from the same fossil layer. Some are exquisitely preserved with details of dark color and body structures. Here, we provide a systematic classification based on new specimens and establish a new mecopteran family.

## 2. Materials and Methods

All specimens were collected from black shales of the lower part of the Yinping Formation near Houdong Village, Sanbing Township, Chaohu City, Anhui Province, China (Figure 1). The Yinping Formation is of the late Capitanian in age [34,35] and it has yielded rich fossils, including sponges, marine bivalves, insects, shrimps, fishes and plants. Fossil insects are diverse, including Orthoptera, Coleoptera, Glosselytrodea, Mecoptera, Hemiptera, Caloneurodea and Megasecoptera [31,36,37,38,39,40,41,42].

The specimens were carefully prepared using a sharp knife. Photographs were taken by a digital camera attached to a Zeiss Discovery V20 microscope (Carl Zeiss AG, Oberkochen, Germany). Most specimens are displayed in two pictures; one was taken in vertical reflected light and immersed under 70% alcohol to improve the contrast of dark color, and the other one was taken in oblique reflected light for displaying veins. Line drawings and maps were made using Adobe Illustrator 2019 graphic software. The specimens are housed in the Nanjing Institute of Geology and Palaeontology, Chinese Academy of Sciences, Nanjing, China.

The wing venation nomenclature generally follows the scheme of Minet et al. [43] and partly follows Bashkuev and Sukatsheva [3]. Venational abbreviations are as follows: C, costa; Sc, subcosta; R_1_, first branch of the radius; Rs, radial sector; M, media; CuA, anterior cubitus, CuP, posterior cubitus; A, anal vein; m-cua, the crossvein between media and anterior cubitus.

**Figure 1 insects-14-00096-f001:**
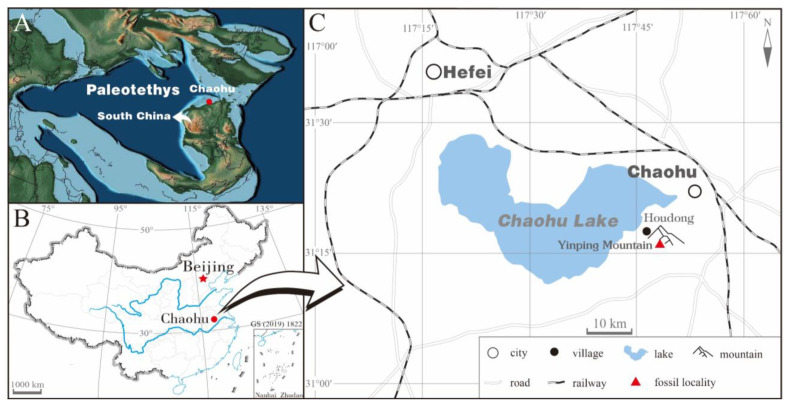
(**A**) Paleomap of the late Capitania, showing the South China Block, base map from Scotese [44]; (**B**) Geographical map of China; (**C**) Geographical map showing the fossil locality.

## 3. Systematic Paleontology

Order Mecoptera Packard, 1886

Family Sinoagetopanorpidae fam. nov.

(ZooBank LSID urn:lsid:zoobank.org:act:D66F0AE9-EB35-4672-86D5-6576ED6D1330)

**Type genus**. *Sinoagetopanorpa* Lin, Nel and Huang, 2010.

**Diagnosis**. Male genitalia clip-shaped, medium-sized; wings broad, oval-shaped, covered with dark color, a hyaline rounded triangular spot at apex of interface between each Rs and M branches (occasionally absent). In forewing, costal area broad, Sc armed with three to four elongated branches that evenly developed; R_1_ more or less curved apically; Rs five-branched, Rs_4_ bifurcated into two branches; M six-branched (occasionally five-branched), M_2_ two-branched, M_4_ two-branched (occasionally single); A_3_ short, occasionally present. In hind wing, costal area broad, Sc single; R_1_ forking into two or three branches; Rs five-branched, Rs_4_ forking into two branches; M five-branched, M_2_ two-branched, M_4_ single. 

**Genera included**. *Sinoagetopanorpa* Lin, Nel and Huang, 2010, *Raragetopanorpa* gen. nov. and *Permoagetopanorpa* gen. nov.

*Sinoagetopanorpa* Lin, Nel and Huang, 2010.

(ZooBank LSID urn:lsid:zoobank.org:act:26971A6F-9533-43B2-B78B-D4D49955346B).

**Type species**. *Sinoagetopanorpa permiana* Lin, Nel and Huang, 2010.

**Other species**. *S. nigra* sp. nov., *S. rotunda* sp. nov., *S. lini* sp. nov., *S. minuta* sp. nov., *S. elegans* sp. nov., *S. grimaldii* sp. nov. and *S. magna* sp. nov.

**Revised diagnosis.** Forewing, Sc with three evenly developed elongated branches; M with six branches, A_3_ present in some cases. Hind wing, R_1_ with two branches or three branches.

*Sinoagetopanorpa permiana* Lin, Nel and Huang, 2010.

(Figure 2, Figure 3 and Figure 19A; ZooBank LSID urn:lsid:zoobank.org:act:B88A8779-1E3F-4D7F-9D25-B42FCEA742C3).

**Type locality and horizon**. Yinping Mountain, Chaohu City, Anhui Province, China; Yinping Formation (Capitanian).

**Material**. Twenty-four specimens, eleven specimens with part and counterpart, and ten complete or nearly so. Holotype, NIGP143428, wing base lacking a small part, with part and counterpart (Figure 2 and Figure 3A,B); from the Yinping Formation, Chaohu City, China (Lin et al. [31], Figure 1, Figure 2, Figure 3 and Figure 4).

Paratypes, NIGP200888 (Figure 3C,D), a complete forewing, with part and counterpart; NIGP200889 (Figure 3E,F), a forewing lacking a small part of apex, with part and counterpart; NIGP200890 (Figure 3G,H), a complete forewing, with part and counterpart. Other unillustrated specimens: NIGP200891, NIGP200892, NIGP200893, NIGP200894, NIGP200895, NIGP200896, NIGP200897, NIGP200898, NIGP200899, NIGP200900, NIGP200901, NIGP200902, NIGP200903, NIGP200904, NIGP200905, NIGP200906, NIGP200907, NIGP200908, NIGP200909, NIGP200910 and NIGP200911.

**Revised diagnosis**. Forewing moderate in size, covered with dense dark-colored spots, a hyaline rounded triangular spot at apex of each interface between Rs and M branches.

**Revised description**. Forewing length 5.6–7.9 mm, width 2.9–4.0 mm, L/W 1.9–2.2; apex somewhat rounded; dark color lessening from wing apex toward wing base, a large patch of dark color at apical wing, basal wing with numerous dark-colored spots, costal area with 5–6 hyaline patches that devoid of dark color, a hyaline rounded triangular spot that devoid of dark color at apex of each interface between Rs and M branches; Sc terminated at about 4/5 of wing, with three evenly developed elongated branches; R_1_ single, curved slightly apically; pterostigma large, with almost a half below the apical R_1_; Rs five-branched, Rs_4_ bifurcated into two branches, Rs_1_ slightly curved upwards, Rs_1+2_ fork does not reach the level of M_2a+b_ fork; a crossvein connected near Rs_1+2_ fork and Rs_3_, stem Rs_3+4_ and stem M_1+2_, Rs_4b_ and M_1_, respectively; M six-branched, M_2_ bifurcated into two branches; the forking patterns of the three branches of M_3+4_ variable; most specimens with M_3_ single and M_4_ two-branched, some with M_3_ two-branched and M_4_ single, and the other with the three branches of M_3+4_ forking at one point; stem M_1+2_ equal to or longer than stem M_3+4_, a crossvein connecting stems M_2_ and M_3_, crossvein m-cua connecting stem M_4_ (or posterior branch of M_4_) and CuA; CuA curved apically, CuP single, straight; the crossvein between basal CuA and CuP oblique to horizontal; the holotypic specimen with R_1_ + Rs fork proximal to the first fork of Sc and M + CuA fork, M + CuA fork slightly proximal to the first fork of Sc; most paratypic specimens with M + CuA fork proximal to R_1_ + Rs fork and R_1_ + Rs fork proximal to the first fork of Sc; two anal veins visible, a crossvein connecting each other near base.

**Remarks**. The holotype is slightly deformed, so the original line drawings (see Lin et al. [31]: Figure 3 and Figure 4) are inaccurate. The upper branch of CuA in the original line drawing is in fact the lower branch of M_4_ (M_4b_).

*Sinoagetopanorpa nigra* sp. nov.

(Figure 4 and Figure 19B; ZooBank LSID urn:lsid:zoobank.org:act:CAF12387-A3EA-4355-AF15-AAAA840CE460).

**Etymology**. The specific name is derived from the Latin word *nigra*, dark, referring to the dark-colored wing.

**Type locality and horizon**. Yinping Mountain, Chaohu City, Anhui Province, China; Yinping Formation (Capitanian).

**Diagnosis**. Forewing almost fully with dark colors, absence of dark-colored spots, absence of a hyaline rounded triangular spot at apex of each interface between Rs and M branches; R_1+2_ fork near the level of M_2a+b_ fork.

**Material**. Holotype, NIGP200912, a complete forewing, with part and counterpart (Figure 4 and Figure 19B). 

**Description**. Forewing length 5.6 mm, width 3.0 mm, W/L 1.9; apex rounded and base slightly shrunken; wing almost fully covered with dark color, with three small hyaline patches at costal area and seven small hyaline spots at wing base; Sc terminated at 2/3 of wing, apical Sc_3_ extending with the same direction as basal stem Sc; R_1_ moderately curved apically; Rs five-branched, stem Rs_1+2_ length 1.2 mm, stem Rs_3+4_ length 0.3 mm, stem Rs_4a+b_ shorter than its branches; a crossvein connecting Rs_1+2_ fork and Rs_3_; Rs merged with R_1_ at a distance of 1.5 mm from wing base; a crossvein connecting stem Rs_3+4_ and stem M_1+2_, Rs_4b_ and M_1_, respectively; M with six branches, stem M_1+2_ as long as stem M_3+4_; both M_2_ and M_4_ bifurcated into two branches; stem M_4_ length 0.2 mm, a crossvein connecting stem M_2_ and M_3_, crossvein m-cua connecting basal M_4b_ and CuA; CuA curved after the connection with corossvein m-cua, the crossvein between basal CuA and CuP oblique; CuP straight, apically curved; M + CuA fork proximal to R_1_ + Rs fork, R_1_ + Rs fork proximal to the first fork of Sc; two anal veins visible, a crossvein connecting A_1_ and A_2_ near base.

*Sinoagetopanorpa rotunda* sp. nov.

(Figure 5 and Figure 19C; ZooBank LSID urn:lsid:zoobank.org:act:9E2C0D32-5E2C-44E1-B69D-FDD9907493C6).

**Etymology**. The specific name is derived from the Latin *rotunda*, rounded, indicating the rounded wing shape.

**Type locality and horizon**. Yinping Mountain, Chaohu City, Anhui Province, China; Yinping Formation (Capitanian).

**Diagnosis**. Forewing distinctly oval-shaped, absence of dark-colored spots; a hyaline rounded triangular spot at apex of interface between each Rs and M branches; R_1_ distinctly curved apically.

**Material**. Three specimens. Holotype, NIGP200913, a complete forewing, with part and counterpart (Figure 5A,B and Figure 19C). Paratypes, NIGP200914, a complete forewing, with part and counterpart (Figure 5C,D,G); NIGP200915, a forewing lacking base, with part and counterpart (Figure 5E,F,H).

**Description**. Forewing broad, broadest area at the 4/5 of wing, length 7.1 mm, width 4.1 mm, L/W 1.7; some elongated hyaline patches developed at anterior and basal wing, and a hyaline rounded triangular spot at apex of each interface between Rs and M branches; crossveins usually accompanied with a small hyaline patch; pterostigma large, with almost a half below the apical R_1_; Sc terminated at 2/3 of wing, with three evenly developed elongated branches; R_1_ single, curved distinctly in the pterostigma; Rs five-branched, Rs_1+2_ fork proximal to M_2a+b_ fork; stem Rs_1+2_ 4 times as long as stem Rs_3+4_; Rs_1_ curved upwards in middle; stem Rs_4a+b_ as long as its branches; a crossvein connecting basal Rs_2_ and Rs_3_; M six-branched, M_2_ and M_4_ forking into two branches; stem M_1+2_ slightly longer than M_3+4_ and as long as stem M_2a+b_; basal Rs_4a_ and M_1_, stem Rs_3+4_ and stem M_1+2_, stem M_2a+b_ and M_3_ connected by a crossvein, respectively, crossvein m-cua connecting basal M_4b_ and CuA; CuA and CuP simple, CuA curved after the crossvein m-cua; the crossvein between CuA and CuP curved; R_1_ + Rs fork and M + CuA fork nearly at the same level and proximal to the first fork of Sc; A_1_ and A_2_ single, a crossvein connecting each other near base. 

**Paratypes**. NIGP200914, a complete forewing, with part and counterpart (Figure 5C,D,G), length 6.9 mm, width 4.0 mm, L/W ratio 1.7; lower margin in the middle concaved obviously; stem Rs_1+2_ twice as long as stem Rs_3+4_; stem M_1+2_ slightly longer than M_3+4_ and nearly twice as long as stem M_2a+b_; M_3_ two-branched, M_4_ single, stem M_3a+b_ short; crossvein m-cua connecting CuA with basal M_4_.

NIGP200915 (Figure 5E,F,H), a forewing with wing base not preserved, with part and counterpart, length 6.7 mm (as preserved), width 3.9 mm; stem Rs_1+2_ 3 times as long as stem Rs_3+4_; stem M_1+2_ as long as stem M_3+4_ and nearly twice as long as stem M_2a+b_; M_3_ two-branched, M_4_ single, stem M_3a+b_ distinctly short; crossvein m-cua connecting CuA and basal M_4_.

*Sinoagetopanorpa lini* sp. nov.

(Figure 6, Figure 7, Figure 8 and Figure 19D; ZooBank LSID urn:lsid:zoobank.org:act:CFDDAB8C-8A18-4A32-AC84-331AE5E360B9).

**Etymology**. The specific name is in honor of the late paleoentomologist, Prof. Qibin Lin, for his extraordinary contribution to the paleoentomology of the Yinping Formation.

**Type locality and horizon**. Yinping Mountain, Chaohu City, Anhui Province, China; Yinping Formation (Capitanian).

**Diagnosis**. Wing without dark-colored spots; a hyaline rounded triangular spot at apex of interface between each Rs and M branches; forewing with Rs_1+2_ fork proximal to M_2a+b_ fork; hind wing devoid of dark color at wing base; R_1_ bifurcating into two terminal branches.

**Material**. Four specimens, two with part and counterpart. Holotype, sex unknown, NIGP200916, with part and counterpart (Figure 6 and Figure 7), a specimen preserved with part of body and four wings, abdomen segments detected; left forewing and right hind wing overlap with each other, left forewing upturned. Paratypes, NIGP200917, a complete forewing, with part and counterpart (Figure 8 and Figure 19D); other unillustrated specimens: NIGP200918 and NIGP200919.

**Description**. Body length 5.3 mm (as preserved), head and thorax preserved obscurely, thorax medium size; abdomen with seven clearly discernible segments, length shorter than width, last two segments narrowed; genitalia relatively small compared with abdomen, clip-shaped, basistyles and dististyles lacking details; the left middle leg preserved with femur, tibia and tarsus; femur robust, length 1.5 mm (as preserved); tibia length 1.9 mm; five tarsi segments preserved (with total length 1.5 mm); tarsomeres gradually shorten from base towards apex; two front legs and left hind leg partly preserved.

Forewing relatively elongated, broadest at 2/3 of wing, right forewing length 7.5 mm, width 3.5 mm, L/W 2.1 (Figure 6E,F and Figure 7C); three small hyaline patches located at costal area; R_1_ single and curved apically; Rs five-branched, a terminal fork visible at right forewing of holotype (Figure 7C), left forewing (Figure 7B) lacking the terminal fork; stem Rs_1+2_ about twice as long as stem Rs_3+4_, Rs_1+2_ fork proximal to M_1+2_ fork, a crossvein connecting basal Rs_2_ and Rs_3_, basal Rs_4b_ and M_1_, stem Rs_3+4_ and stem M_1+2_, respectively; M six-branched, stem M_1+2_ 1.5 times as along as stem M_3+4_, both M_2_ and M_4_ two-branched; a crossvein connecting stem M_2a+b_ and M_3_, crossvein m-cua connecting CuA with M near M_4a+b_ fork; CuA curved after the crossvein m-cua, CuP single, straight; the crossvein connecting basal CuA and CuP nearly horizontal; R_1_ + Rs fork at the same level as M + CuA fork and proximal to the first fork of Sc; two anal veins visible, a crossvein connecting A_1_ and A_2_.

Hind wing smaller than forewing; left hind wing length 6.4 mm, width 3.2 mm, L/W 2.0 (Figure 6G,H and Figure 7D,E); wing with dense dark color, a large hyaline patch stretched from the middle part of wing base to middle wing, a hyaline rounded triangular spot at apex of interface between each Rs and M branches; costal area broad, Sc single, terminated at middle wing, curved apically; R_1_ bifurcated into two branches at middle wing; Rs five-branched, stem Rs_1+2_ more than 3 times as long as stem Rs_3+4_; stem Rs_4a+b_ longer than its branches, Rs_1+2_ forking at the same level as M_1+2_ fork; a crossvein between R_1b_ and Rs_1_, basal Rs_2_ and Rs_3_, Rs_3_ and Rs_4a_, respectively, and the crossveins inside the small spots devoid of dark color; M five-branched, M_2_ two-branched, stem M_1+2_ as long as stem M_3+4_ and M_2a+b_; basal stem Rs_4a+b_ and stem M_1+2_, Rs_4b_ and M_1_, stem M_2a+b_ and M_3_ connected by a crossvein, respectively; crossvein m-cua connecting M_3+4_ fork and CuA; CuA and CuP straight, the crossvein between basal CuA and CuP oblique; left forewing with R_1_ + Rs fork slightly proximal to M + CuA fork; right forewing with R_1_ + Rs fork distinctly proximal to M + CuA fork; two straight anal veins visible. 

Paratypes: NIGP200917 (Figure 8 and Figure 19D), forewing, length 6.4 mm, width 3.0 mm, W/L 2.1; three relatively large irregular hyaline patches covered at the area of costa, subcosta and along the R_1_, a hyaline rounded triangular spot at apex of each interface between Rs_2–4_ and M branches, some crossveins in spots and approximately 10 small spots at wing base; stem M_1+2_ slightly longer than stem M_3+4_ and twice as long as stem M_2a+b_; the three branches of M_3+4_ nearly forking at one point.

**Remarks**. The dark color inside the forewings of the holotype is too poorly preserved to reconstruct. 

*Sinoagetopanorpa minuta* sp. nov.

(Figure 9 and Figure 19E; ZooBank LSID urn:lsid:zoobank.org:act:512A5737-065F-49D8-BDFB-68F002A18129).

**Etymology**. The specific name is derived from the Latin *minuta*, small, indicating the small wing size.

**Type locality and horizon**. Yinping Mountain, Chaohu City, Anhui Province, China; Yinping Formation (Capitanian).

**Diagnosis**. Forewing small, a hyaline rounded triangular spot at end of each interface of Rs and M branches; Rs_4_ forking late, A_3_ short, the area between A_3_ and wing margin narrow.

**Material**. Only holotype (NIGP200920) examined, a nearly complete forewing, with part and counterpart (Figure 9 and Figure 19E).

**Description**. Forewing with apex rounded and broad, length 4.8 mm, width 2.6 mm, L/W 1.9; wing with dark color, more than 10 irregular hyaline patches focused on anterior and basal wing, some crossveins accompanied with small hyaline patches, a hyaline rounded triangular spot at apex of each interface of Rs and M_1–3_ branches; Sc terminated at a distance of 2/3 from wing base; R_1_ single and nearly straight; Rs five-branched, stem Rs_1+2_ length 0.6 mm, stem Rs_3+4_ length 0.3 mm; Rs_1+2_ forking at the same level as M_1+2_ fork; Rs_4_ forking apically; stem Rs_4a+b_ twice as long as its branches; a crossvein connecting basal Rs_2_ and Rs_3_; M six-branched, M_2_ and M_4_ both bifurcated into two branches, stem M_1+2_ as long as stem M_2a+b_ and slightly longer than stem M_3+4_, stem M_4a+b_ very short; Rs_3+4_ fork and stem M_1+2_, Rs_4a+b_ fork and M_1_, stem M_2a+b_ and M_3_ connected by a crossvein, respectively, crossvein m-cua connecting M_4a+b_ fork and CuA; CuA curved after the crossvein m-cua, CuP straight; the crossvein between CuA and CuP robust and nearly horizontal; M + CuA fork proximal to R_1_ + Rs fork, R_1_ + Rs fork slightly proximal to the first fork of Sc; three anal veins visible, A_3_ short, very close to wing margin.

*Sinoagetopanorpa elegans* sp. nov.

(Figure 10, Figure 11 and Figure 19F; ZooBank LSID urn:lsid:zoobank.org:act:7BE01613-3A97-4141-9F94-248339997070).

**Etymology**. The specific epithet is derived from the Latin *elegans*, indicating the well-preserved four wings. 

**Type locality and horizon**. Yinping Mountain, Chaohu City, Anhui Province, China; Yinping Formation (Capitanian).

**Diagnosis**. Forewing with numerous dark-colored spots, base narrow, a hyaline rounded triangular spot at apex of each interface of Rs and M branches; Rs_4a+b_ forked late, A_3_ distinctly developed, the area between A_3_ and wing margin broad. Hind wing with Rs_4a+b_ forked very late, R_1_ with three terminal branches.

**Material**. Holotype, NIGP200921 (Figure 10 and Figure 11), two forewings and two hind wings preserved in one specimen, interpreted as one individual. One forewing is well-preserved, the other forewing lacking apex; two hind wings overlap with each other, poorly preserved.

**Description**. Forewing with apex rounded, base obviously shrunken, the complete forewing length 5.6 mm, width 2.6 mm, L/M 2.2; dark color denser at the apical wing than basal wing, costal area with 5–6 hyaline patches; Sc terminated at a distance of 3.9 mm from wing base; R_1_ single, smoothly curved near apex; pterostigma large, half below the apical R_1_; Rs five-branched, Rs_4_ bifurcated into two terminal branches, Rs_1_ curved upwards, Rs_1+2_ fork at the same level as M_1+2_ fork, stem Rs_1+2_ 1.5 times as long as stem Rs_3+4_, stem Rs_4a+b_ long and 1.5 times as long as its branches; M six-branched, M_2_ bifurcating into two branches, stem M_1+2_ about 1.5–1.7 times as long as stem M_3+4_, the three branches of M_3+4_ forking close, resulting in one forewing with M_3_ two-branched and M_4_ single, the other forewing with M_3_ single and M_4_ two-branched; a crossvein connecting stem Rs_3+4_ and M_1+2_, stem M_2a+b_ and M_3_, respectively, crossvein m-cua connecting basal M_4_ (or M_4b_) and CuA; CuA curved apically, CuP straight and single; M + CuA proximal to R_1_ + Rs, R_1_ + Rs proximal to the first fork of Sc; three anal veins, A_3_ short, the area between A_3_ and wing margin broad.

Hind wing poorly preserved with dark color; costal area broad, ca. 3 times as wide as subcostal area; Sc abruptly curved to costa apically; R_1_ forking near the same level as apical Sc; Rs five-branched, Rs_4_ bifurcated into two branches, one hind wing with stem Rs_1+2_ slightly longer than Rs_3+4_, the other hind wing with Rs_1+2_ nearly twice as long as stem Rs_3+4_; Rs_4a+b_ fork distinctly distad to M_2a+b_ fork; a crossvein connecting stem Rs_3+4_ and stem M_1+2_; M five-branched, M_2_ with two branches, stem M_1+2_ longer than stem M_3+4_; CuA and CuP single and straight, anal veins absent.

*Sinoagetopanorpa grimaldii* sp. nov.

(Figure 12 and Figure 19G; ZooBank LSID urn:lsid:zoobank.org:act:40BB5B06-8910-4165-8253-B7D1591D0E16).

**Etymology**. The specific name is dedicated to the famous American paleoentomologist David Grimaldi.

**Type locality and horizon**. Yinping Mountain, Chaohu City, Anhui Province, China; Yinping Formation (Capitanian).

**Diagnosis**. Forewing with two distinct dark stripes across the wing, with some dark-colored spots, apex of each Rs and M branches covered with a dark-colored spots and absence of the hyaline triangular spot.

**Material**. Four specimens, two of them with part and counterpart. Holotype, NIGP200922, a complete forewing (Figure 12A,B and Figure 19G). Paratypes, NIGP200923, a forewing lacking apex (Figure 12C,D); NIGP200924, a forewing lacking wing base with part and counterpart; NIGP200925, lacking a small part of wing apex, with part and counterpart.

**Description**. Holotype, NIGP200922, forewing length 8.0 mm, width 4.0 mm, W/L 2.0, widest at middle wing, obviously tapering to base and apex; two dark stripes vertically lined across the wing, the larger one located at apex of R_1_, the smaller one located at Sc_3_; many dark-colored spots scattering at wing apex, middle and basal wing, but not combined into stripe; Sc terminated at 3/4 of wing; R_1_ curved apically; Rs five-branched, stem Rs_1+2_ twice as long as stem Rs_3+4_, Rs_1+2_ forking at the same level as M_1+2_ fork, stem Rs_4a+b_ shorter than its branches; M six-branched, stem M_1+2_ slightly longer than M_3+4_ and M_2a+b_; M_3_ two-branched, M_4_ single, stem M_3_ short; a crossvein connecting Rs_4b_ and M_1_, stem M_1+2_ and basal Rs_4a+b_, stem M_2a+b_ and M_3a_, respectively; crossvein m-cua connecting basal M_4_ and CuA; CuA single, curved apically, merged with M at the level of first Sc fork; CuP single, apically curved, the crossvein between basal CuA and CuP oblique; R_1_ + Rs fork distinctly proximal to M + CuA and the first fork of Sc; two long anal veins, A_1_ leaned to CuP apically, A_2_ terminated at 1/3 of wing, a crossvein connecting A_1_ and A_2_.

Paratypes: NIGP200923 (Figure 12C,D), forewing length 6.9 mm (as preserved), width 3.5 mm; stem M_1+2_ as long as stem M_3+4_, the three branches of M_3+4_ forking at one point. NIGP200924, forewing length 6.9 mm (as preserved), width 3.6 mm; stem Rs_1+2_ 3 times as long as stem Rs_3+4_; M_3_ single, M_4_ two-branched, stem M_1+2_ distinctly longer than stem M_3+4_. NIGP200925; forewing length 8.3 mm (as preserved), width 3.9 mm, deformed and overlapped by another wing fragment; M_3_ single, M_4_ two-branched, stem M_1+2_ slightly longer than stem M_3+4_.

*Sinoagetopanorpa magna* sp. nov.

(Figure 13 and Figure 19H; ZooBank LSID urn:lsid:zoobank.org:act:D309ED60-3469-4987-AEBA-56BCFF4B615A).

**Etymology**. The specific name is derived from the Latin word *magna*, large, indicating the large-sized wing. 

**Type locality and horizon**. Yinping Mountain, Chaohu City, Anhui Province, China; Yinping Formation (Capitanian).

**Diagnosis**. Forewing relatively large, with two dark stripes, numerous small dark-colored spots at sides of the stripes; a distinct crossvein connecting base of Rs_1_ and R_1_.

**Material**. Holotype NIGP200926, with part and counterpart (Figure 13 and Figure 19H), with veins well-preserved but lacking wing apex and part of base.

**Description.** Forewing length 8.2 mm (as preserved), estimated length 10.0 mm, width 5.0 mm, with two distinct colored stripes vertically lined across the wing, one located at apex of R_1_ and tapering to posterior wing, the other one located at Sc_3_; numerous small dark-colored spots scattering at sides of the stripes, each spot apart from the others; Sc terminated at a distance of 7.5 mm from wing base; R_1_ curved apically; Rs five-branched, stem Rs_1+2_ length 1.3 mm, stem Rs_3+4_ length 0.8 mm, stem Rs_4a+b_ length 2.2 mm, Rs_1+2_ fork slightly proximal to M_1+2_ fork, a crossvein connecting basal Rs_1_ and R_1_; M six-branched, stem M_1+2_ length 1.4 mm, stem M_2a+b_ length 1.1 mm, stem M_3+4_ length 1.0 mm, M_3+4_ three-branched and forking at one point; stem M_1+2_ and Rs_3+4_ fork, stem M_2a+b_ and upper branch of M_3+4_ connected by a crossvein, respectively; crossvein m-cua connecting lower branch of M_3+4_ and CuA; CuA curved after the crossvein m-cua, CuP single and curved apically; the crossvein between basal CuA and CuP robust and nearly horizontal; two anal veins detected, A_1_ leaned to CuP near apex; A_2_ terminated at a distance of 4.3 mm from wing base. 

*Raragetopanorpa* gen. nov.

(ZooBank LSID urn:lsid:zoobank.org:act:61003EF7-F803-4261-9AE0-6CF9B65FD5B9).

**Etymology**. The generic name combines the Latin word *rara*, rare, indicating only one specimen has been found, and a mecopteran generic name *Agetopanorpa*.

**Diagnosis**. Sc with three evenly developed elongated branches; M five-branched with M_2_ two-branched and M_4_ single.

**Type species.***Raragetopanorpa zhangi* gen. et sp. nov., genus monotypic.

*Raragetopanorpa zhangi* sp. nov.

(Figure 14 and Figure 19I; ZooBank LSID urn:lsid:zoobank.org:act:54B79578-0A41-4D0C-9983-6D52E4723AFD).

**Etymology**. The species name *zhangi* is in honor of the late paleoentomologist, Prof. Junfeng Zhang, for his contribution to Chinese paleoentomology. 

**Type locality and horizon**. Yinping Mountain, Chaohu City, Anhui Province, China; Yinping Formation (Capitanian).

**Diagnosis**. As for the genus.

**Material**. Holotype, NIGP200927, a nearly complete forewing with base poorly preserved (Figure 14 and Figure 19I). 

**Description**. Forewing with apex somewhat rounded, length 6.4 mm, width 3.3 mm, L/W 1.9; wing with dark color, several distinct dark-colored spots at top of middle wing, costal area with four hyaline spots, a hyaline rounded triangular spot at apex of each Rs and M branches, some crossveins inside a small spot; Sc terminated distad to the level of Rs_4a+b_ fork; R_1_ single, straight, smoothly curved apically; Rs five-branched, Rs_1_ curved, stem Rs_1+2_ length 1.0 mm, stem Rs_3+4_ length 0.3 mm, stem Rs_4a+b_ as long as its branches; M five-branched, stem M_1+2_ slightly longer than stem M_3+4_; stem M_2a+b_ as long as stem M_3+4_; basal Rs_4a+b_ and M_1+2_, basal stem Rs_4_ and M_1_, middle of M_2a+b_ and M_3_ connected by a crossvein, respectively, crossvein m-cua connecting basal M_4_ and CuA; CuA curved after crossvein m-cua, merged with M at the same level as R_1_ + Rs fork and proximal to the first fork of Sc; CuP single and straight; the crossvein between basal CuA and CuP oblique; two anal veins detected, A_1_ curved at basal half, A_2_ shorter and curved downwards at apex.

*Permoagetopanorpa* gen. nov.

(ZooBank LSID urn:lsid:zoobank.org:act:C1F1E1AB-A4B2-42DA-A8EF-E225C91641BE).

**Etymology**. The generic name combines *Permo*, the Permian period, and mecopteran generic name *Agetopanorpa*.

**Diagnosis**. Forewing, Sc with four evenly developed elongated branches; M six-branched, with M_2_ and M_4_ bifurcated into two branches.

**Type species**. *Permoagetopanorpa yinpingensis* gen. et sp. nov.

**Other species**. *Permoagetopanorpa incompleta* sp. nov.

*Permoagetopanorpa yinpingensis* sp. nov.

(Figure 15 and Figure 19J; ZooBank LSID urn:lsid:zoobank.org:act:344ED4C6-A618-4729-B6BC-C77F22D9976A).

**Etymology**. The species name is derived from the Yinping Formation, where the specimen was collected.

**Type locality and horizon**. Yinping Mountain, Chaohu City, Anhui Province, China; Yinping Formation (Capitanian).

**Diagnosis**. Forewing moderately large in this group, Sc branches long, Sc_3_, Sc_4_, and stem Sc forming relatively small angles. 

**Material**. Holotype, NIGP200928, a specimen lacking some part of apex, with part and counterpart (Figure 15 and Figure 19J). 

**Description**. Forewing relatively broad, length 8.0 mm, width 4.3 mm, broadest at 2/3 of wing; wing with dense dark color, five hyaline patches at costal area, a hyaline rounded triangular spot at each apex between Rs_1_, Rs_2_ and Rs_3_; Sc terminated at a distance of 5.9 mm from wing base, the first fork of Sc at a distance of 2.3 mm from wing base; R_1_ single and curved apically; Rs five-branched, stem Rs_1+2_ length 1.2 mm, stem Rs_3+4_ length 0.3 mm, stem Rs_4a+b_ length 1.8 mm, Rs_1+2_ fork distad to M_1+2_ fork, Rs_1_ curved upwards, basal Rs_2_ and Rs_3_ connected by a crossvein; M six-branched, M_2_ and M_4_ bifurcated into two branches, stem M_1+2_ length 1.3 mm, stem M_3+4_ length 0.9 mm; stem Rs_3+4_ and stem M_1+2_, stem M_2a+b_ and M_3_ connected by a crossvein, respectively; the crossvein m-cua indistinct, inferred from the sharply curved M_4a+b_; CuA curved at apical part; CuA merged with M at the same level as R_1_ + Rs fork and proximal to the first fork of Sc; CuP straight; two anal veins detected; a crossvein connecting A_1_ and A_2_.

*Permoagetopanorpa incompleta* sp. nov.

(Figure 16 and Figure 19K; ZooBank LSID urn:lsid:zoobank.org:act:420F6ACA-BA85-49ED-8A51-57B2C5299F4C).

**Etymology**. The specific name is derived from the incompletely preserved forewing.

**Type locality and horizon**. Yinping Mountain, Chaohu City, Anhui Province, China; Yinping Formation (Capitanian).

**Diagnosis**. Forewing relatively small, Sc branches relatively short, Sc_3_, Sc_4_ and Sc stem forming relatively large angles.

**Material**. Holotype, NIGP200929, lacking a large part of wing base, overlapped by other wing fragment at anal area, with part and counterpart (Figure 16 and Figure 19K).

**Description**. Forewing covered with dense dark color, wing length 5.2 mm (as preserved), width 3.1 mm, with some small hyaline patches at costal area; a hyaline rounded triangular spot at apex of each interface between Rs and M_1–3_ branches; Sc_3_, Sc_4_ and Sc stem forming nearly 60-degree angles, Sc_1_ and Sc_2_ forming a 30-degree angle, each branch of Sc nearly the same length; R_1_ single and curved apically; Rs five-branched, stem Rs_1+2_ length 0.7 mm, stem Rs_3+4_ length 0.5 mm, Rs_4a+b_ longer than its branches, Rs_1+2_ forking at the same level as M_1+2_ fork; a possible crossvein inside a spot connecting basal Rs_2_ and Rs_3_; M six-branched, with M_2_ and M_4_ forking into two branches, stem M_1+2_ length 0.9 mm, stem M_3+4_ length 0.7 mm, stem M_2a+b_ length 0.8 mm, stem M_4a+b_ short; a crossvein connecting stem Rs_3+4_ and stem M_1+2_, stem M_2a+b_ and M_3_; CuA curved at the apical part; other veins not preserved.


**Hind wings of Sinoagetopanorpidae fam. nov.**


The hind wings of Sinoageotpanorpidae fam. nov. are often preserved as isolated wings. Fourteen isolated hind wings (NIGP200930, NIGP200931, NIGP200932, NIGP200933, NIGP200934, NIGP200935, NIGP200936, NIGP200937, NIGP200938, NIGP200939, NIGP200940, NIGP200941, NIGP200942 and NIGP200943) were found (four with part and counterpart; NIGP200932 preserved a pair of hind wings). Two hind wings accompanied with forewings (*S. lini*. sp. nov. and *S. elegans* sp. nov.), and the rest of hind wings remain elusive in terms of their systematic positions. Similar to forewings, the hind wings are very stable in venation, with two types of venational patterns distinguished. The most common one (13/14) with three-branched R_1_ is represented by *S. elegans* sp. nov., but the hind wings of *S. elegans* sp. nov. are too incomplete with poor preservation to compare with other isolated hind wings; thus, no isolated hind wings are reluctantly attributed to *S. elegans* sp. nov. The rare one (one specimen) with two-branched R_1_ is represented by *S. lini* sp. nov.

The specimen NIGP200930 (Figure 17A,B and Figure 18A) is a well-preserved hind wing and described as follows: wing broad, base narrow, broadest at near middle wing, length 6.6 mm, width 3.6 mm, L/W 1.8; wing apex with dark color, costal area with three hyaline patches, basal one oval shaped and large; basal wing devoid of dark color; lines of dark color along longitudinal veins; costal area broad, ca. 3 times as wide as subcostal area; Sc abruptly curved to costa near apex; R_1_ with three evenly developed terminal branches, R_1b_ paralleled to R_1c_, forking near the level of apical Sc; Rs five-branched, with Rs_4_ bifurcated into two branches, stem Rs_1+2_ more than twice as long as stem Rs_3+4_, a crossvein connected near Rs_1+2_ fork and Rs_3_, stem Rs_3+4_ and M_1+2_, Rs_4b_ and M_1_, respectively; M five-branched, M_2_ two-branched, M_4_ single, stem M_1+2_ as long as stem M_3+4_; a crossvein connecting stem M_2a+b_ and M_3_, crossvein m-cua connecting basal M_4_ and CuA; CuA straight, connected with M near wing base; CuP straight, the crossvein between basal CuA and CuP nearly horizontal; A_1_ straight, connected with CuP near wing base; A_2_ with one long branch and one short branch, forking near wing base; a crossvein connecting basal A_2_ and A_1_, A_3_ very short and close to wing margin.


**Key to genera and species of Sinoagetopanorpidae fam. nov.**


Sc with four evenly developed elongated branches……………2 *Permoagetopanorpa* gen. nov.-Sc with three evenly developed elongated branches………………………………3Wing relatively large; Sc branches relatively long; Sc_3_, Sc_4_ and stem Sc forming relatively small angles………………………………*Permoagetopanorpa yinpingensis* sp. nov.-Wing relatively small; Sc branches relatively short; Sc_3_, Sc_4_ and stem Sc forming relatively large angles………………………*Permoagetopanorpa incompleta* sp. nov.M six-branched…………………………………………………………4 *Sinoagetopanorpa*-M five-branched…………………………… *Raragetopanorpa zhangi* gen. et sp. nov.Wing covered with numeral dark-colored spots and some spots fused in stripes; a round dark-colored spot at apex of each Rs and M branches……………………………5-A hyaline rounded triangular spot at apex of each interface between Rs and M branches………………………………………………………………………………6-Absence of the hyaline rounded triangular spot at apex of each interface between Rs and M branches; R_1+2_ fork near the level of M_2a+b_ fork……… *Sinoagetopanorpa nigra* sp. nov.Wing moderately large; fewer dark-colored spots at sides of stripes…………*Sinoagetopanorpa grimaldii* sp. nov.-Wing relatively large, more dark-colored spots at sides of stripes; a crossvein connecting base of Rs_1_ and R_1_…………………… *Sinoagetopanorpa magna* sp. nov.A_3_ developed, Rs_4a+b_ forking late……………………………………………………………7-A_3_ absent, Rs_4a+b_ forking early…………………………………………………………8Wing with dense dark color; the area between A_3_ and wing margin narrow………………………………………………………*Sinoagetopanorpa minuta* sp. nov.-Wing with numerous dark-colored spots; the area between A_3_ and wing margin broad; R_1_ with three terminal branches in hind wing ………………………………………………*Sinoagetopanorpa elegans* sp. nov.Wing with dense colored spots…………*Sinoagetopanorpa permiana* Lin, Nel and Huang, 2010Wing with dense dark color; hind wing with R_1_ two-branched………………………*Sinoagetopanorpa lini* sp. nov.Wing broad, with dense dark color; apex of R_1_ distinctly curved………………………………………………… *Sinoagetopanorpa rotunda* sp. nov.

**Figure 19 insects-14-00096-f019:**
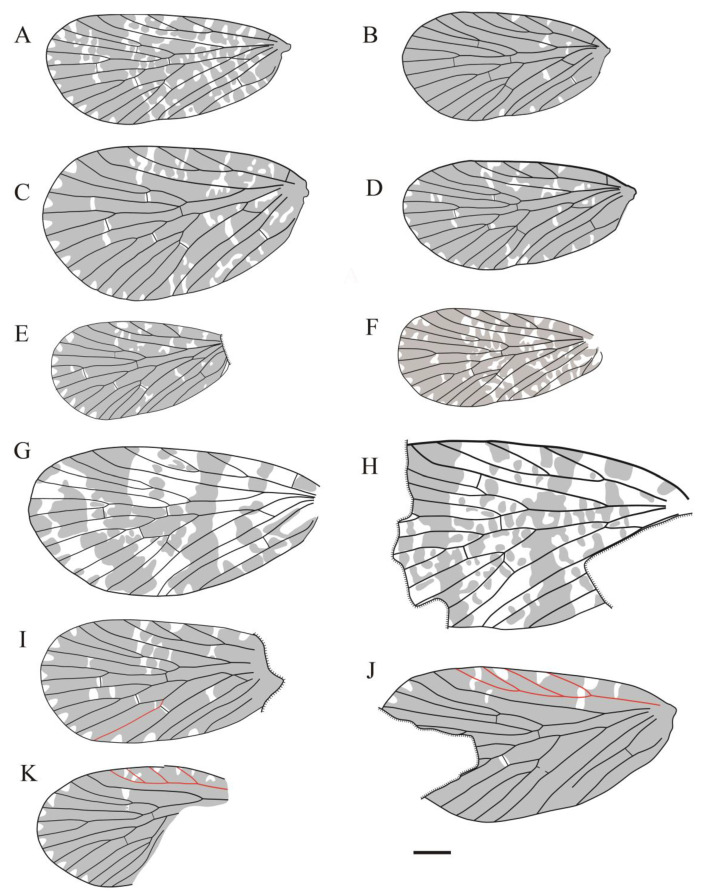
Line drawings of all sinoagetopanorpid species: (**A**) *Sinoagetopanorpa permiana* Lin, Nel and Huang, 2010, NIGP200888 (paratype); (**B**) *S. nigra* sp. nov., NIGP200912 (holotype); (**C**) *S. rotunda* sp. nov., NIGP200913 (holotype); (**D**) *S. lini* sp. nov., NIGP200917 (paratype); (**E**) *S. minuta* sp. nov., NIGP200920 (holotype); (**F**) *S. elegans* sp. nov., NIGP200921 (holotype); (**G**) *S. grimaldii* sp. nov., NIGP200922 (holotype) (mirror image); (**H**) *S. magna* sp. nov., NIGP20926 (holotype); (**I**) *Raragetopanorpa zhangi* sp. nov., NIGP200927 (holotype); (**J**) *Permoagetopanorpa yinpingensis* sp. nov., NIGP200928 (holotype); (**K**) *P. incompleta* sp. nov., NIGP200929 (holotype). Scale bar represents 1 mm.

## 4. Discussion

Sinoagetopanorpidae fam. nov. resemble members of the subfamily Agetopanorpinae of Permochoristidae in venation. Both groups are characterized by the following venational characteristics: Sc usually with three evenly developed elongated branches, Rs with five branches and M generally with six branches. However, it is very conspicuous that sinoagetopanorpids possess broad oval-shaped forewings with broad costal area, and the forking patterns of the three branches of the M_3+4_ forks are varied: M_4_ bifurcating into two branches and M_3_ single or M_3_ bifurcating into two branches and M_4_ single, or even three branches of M_3+4_ forking at the same point. With the consideration of these different forewing characteristics, we erected the new family Sinoagetopanorpidae fam. nov.

Sinoagetopanorpidae resemble Choristopsychidae Martynov, 1937 in having broad oval-shaped wings: Sinoagetopanorpidae possess a wing aspect ratio of 1.7–2.2, while 1.5–2.0 in Choristopsychidae, and both families show a broad costal area with three evenly developed branched Sc. Choristopsychidae was placed by many authors in the Permochoristidae family for its venational similarity with Agetopanorpinae [45,46,47]; however, Qiao et al. [48] discovered numerous exquisite specimens from the Middle−Upper Jurassic (originally assigned to the Middle Jurassic) Daohugou biota, and erected the family Choristopsychidae. Sinoagetopanorpidae differ from Choristopsychidae in a combination of the following characteristics: in terms of forewings, Sinoagetopanorpidae have at least five-branched Rs instead of four-branched, six-branched M with three-branched M_3+4_ instead of five-branched M with two-branched M_3+4_; in addition, the hind wings of Sinoagetopanorpidae have single Sc instead of two-branched Sc; furthermore, Sinoagetopanorpidae are confined to the Permian strata, whereas Choristopsychidae occurred in the Jurassic. 

Based on our study of abundant new specimens, we found that the venational pattern of Sinoagetopanorpidae is rather stable: Sc generally with three branches (except the two new species of *Permoagetopanorpa*); Rs five-branched; M with six branches where M_2_ and M_4_ (or M_3_) bifurcate into two branches (except for *Raragetopanorpa zhangi* sp. nov., which possesses a five-branched M with two-branched M_2_ and a single M_4_) and anal veins generally with two branches, but A_3_ detected in *S. elegans* and *S. minuta*. To distinguish these species, dark-colored pattens play another important role in species-level classification. In the forewings, four kinds of colored patterns are recognized: (1) the absence of a hyaline rounded triangular spot at the apex of each interface between Rs and M branches; (2) the apex of each Rs and M vein has a dark-colored spot and two dark-colored stripes across the wing; (3) a hyaline rounded triangular spot at the apex of each interface between Rs and M branches, the wing with a dense dark color; and (4) a hyaline rounded triangular spot at the apex of each interface between Rs and M branches, the wing with numerous colored spots.

Our discovery of two specimens preserved with both forewings has some implications for venational variations. The holotype of *S. lini* sp. nov. possesses the right forewing with Rs_1_ armed with a terminal fork, while the left forewing with a single Rs_1_. The variation of the terminal fork in the anterior branches of Rs in one individual can be found in other mecopteran families, such as Cimbrophlibiidae [49] and Panorpidae [50,51]. The two forewings of holotype of *S. elegans* sp. nov. show variation in the forking pattern of the three branches of M_3+4_: one forewing with M_3_ two-branched and M_4_ single, but the other with M_4_ two-branched and M_3_ single, indicating that the forking pattern of M_3+4_ is unstable. Therefore, we do not regard these characters as an interspecific diagnostic characteristic.

During the Capitanian, the fossil locality was near 32° N on the Northeast Yangtze Platform in the eastern Paleotethys, which split from the Gondwana supercontinent during the Silurian [44,52], and it possibly had a lagoonal paleoenvironment under the large-scale regression [35,38]. The extant known scorpionflies, with no exception, are weak flyers with low dispersal capacity. Therefore, the endemic mecopteran group Sinoagetopanorpidae fam. nov., the representative mecopterans in the Yinping Formation, might have evolved independently on the Yangtze Platform. The new family from the late Capitanian might have become extinct during the end-Guadalupian mass extinction, which was possibly associated with large-scale volcanic activities [53,54].

## Figures and Tables

**Figure 2 insects-14-00096-f002:**
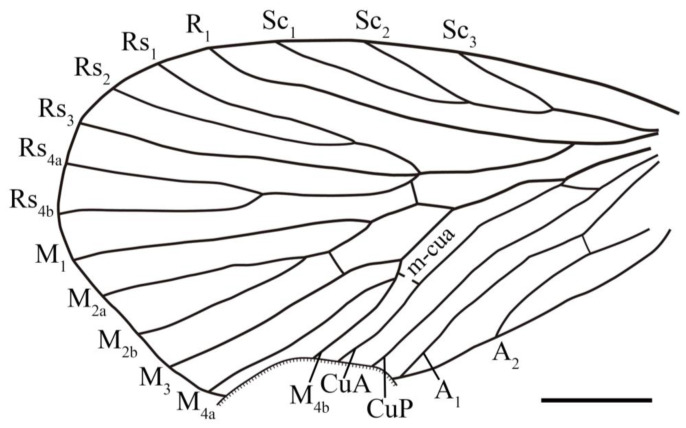
New line drawing of *Sinoagetopanorpa permiana* Lin, Nel and Huang, 2010, based on NIGP143428. Scale bar represents 1 mm.

**Figure 3 insects-14-00096-f003:**
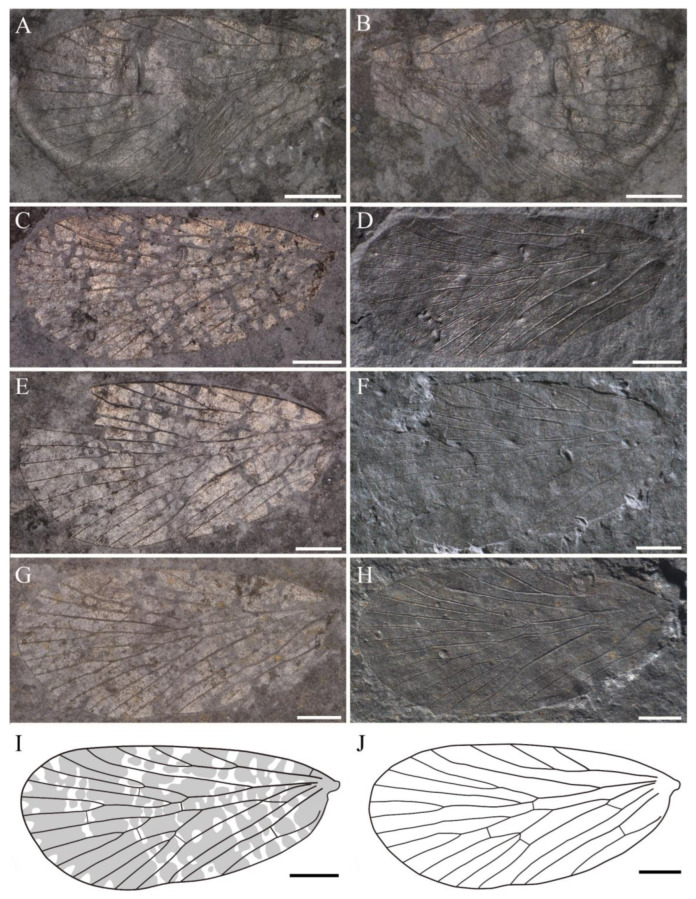
*Sinoagetopanorpa permiana* Lin, Nel and Huang, 2010. (**A**) Photograph of part of NIGP143428 (holotype); (**B**) Photograph of counterpart of NIGP143428 (holotype); (**C**,**D**) Photographs of NIGP200888 (paratype); (**E**,**F**) Photographs of NIGP200889 (paratype) (mirror image); (**G**,**H**) Photographs of NIGP200890 (paratype); (**I**) Line drawing of NIGP200888; (**J**) Line drawing of NIGP200890, dark color not illustrated. (**A**–**C**,**E**,**G**) were taken when specimens were immersed under 70% alcohol in vertical reflected light; (**D**,**F**,**H**) were taken in oblique reflected light. Scale bars represent 1 mm in (**A**–**J**).

**Figure 4 insects-14-00096-f004:**
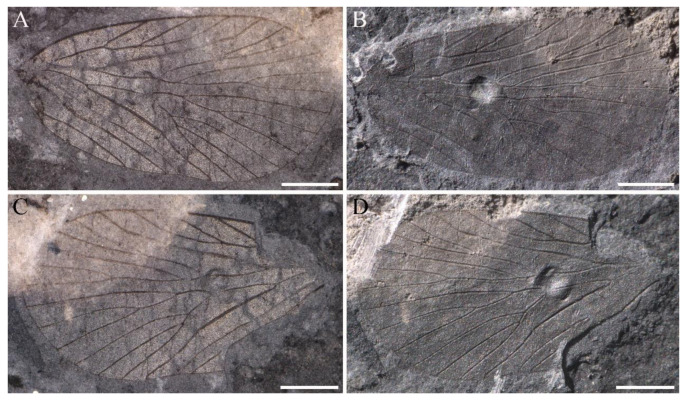
*Sinoagetopanorpa nigra* sp. nov., NIGP200912 (holotype). (**A**,**B**) Photographs of part; (**C**,**D**) Photographs of counterpart; (**A**,**C**) were taken when specimens were immersed under 70% alcohol in vertical reflected light; (**B**,**D**) were taken in oblique reflected light. Scale bars represent 1 mm in (**A**–**D**).

**Figure 5 insects-14-00096-f005:**
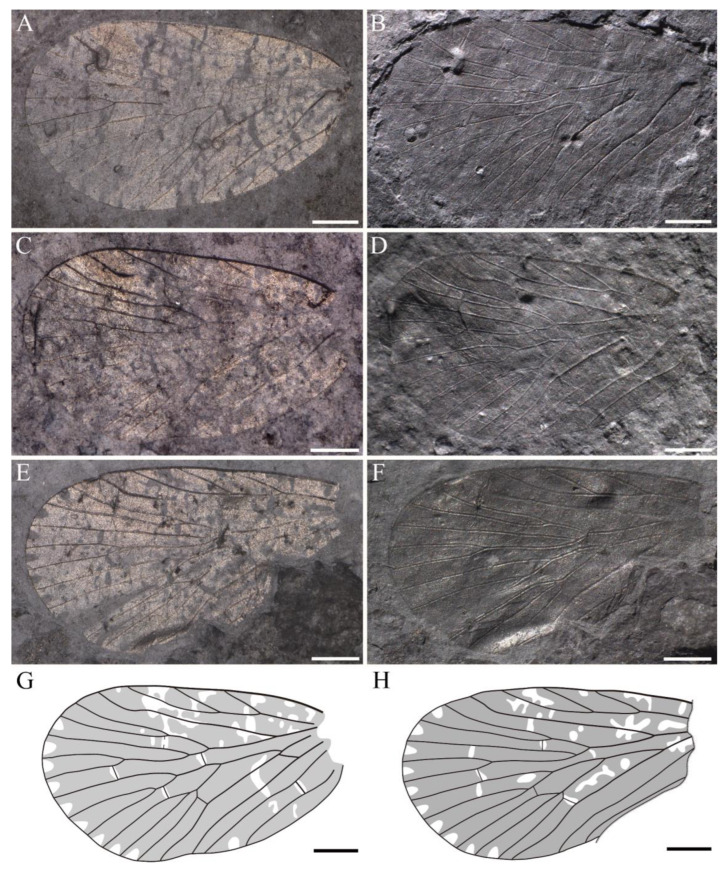
*Sinoagetopanorpa rotunda* sp. nov. (**A**,**B**) Photographs of NIGP200913 (holotype); (**C**,**D**) Photographs of NIGP200914 (paratype) (mirror image); (**E**,**F**) Photographs of NIGP200915 (paratype); (**G**) Line drawing of NIPG200914; (**H**) Line drawing of NIPG200915; (**A**,**C**,**E**) were taken when specimens were immersed under 70% alcohol in vertical reflected light; (**B**,**D**,**F**) were taken in oblique reflected light. Scale bars represent 1 mm in (**A**–**F**).

**Figure 6 insects-14-00096-f006:**
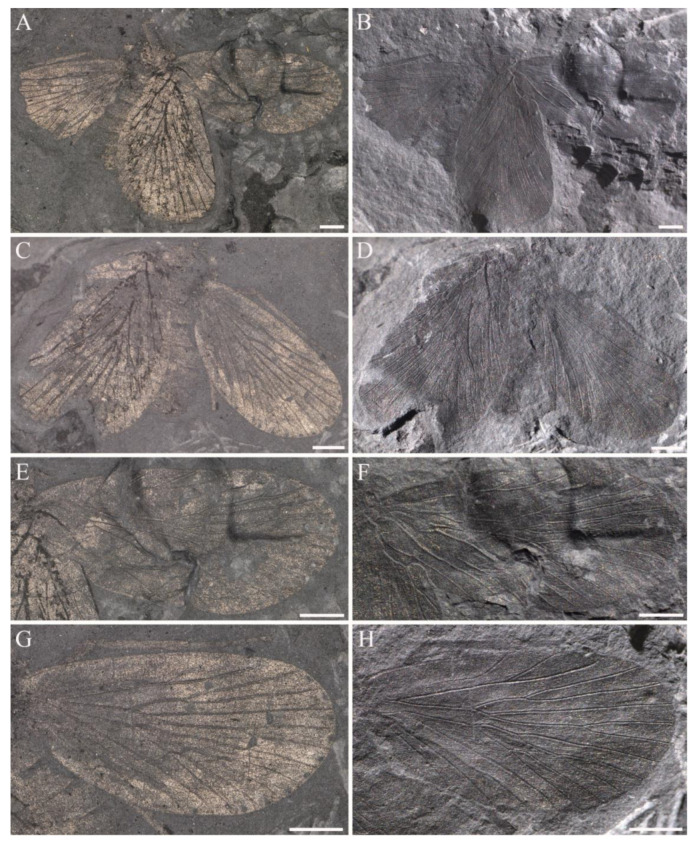
*Sinoagetopanorpa lini* sp. nov., NIGP200916 (holotype). (**A**,**B**) Photographs of part, showing general habitus; (**C**,**D**) Photographs of counterpart, showing general habitus; (**E**,**F**) Photographs of right forewing, enlargement from (**A**,**B**); (**G**,**H**) Photographs of left hind wing, enlargement from (**C**,**D**); (**A**,**C**,**E**,**G**) were taken when specimens were immersed under 70% alcohol in vertical reflected light; (**B**,**D**,**F**,**H**) were taken in oblique reflected light. Scale bars represent 1 mm in (**A**–**H**).

**Figure 7 insects-14-00096-f007:**
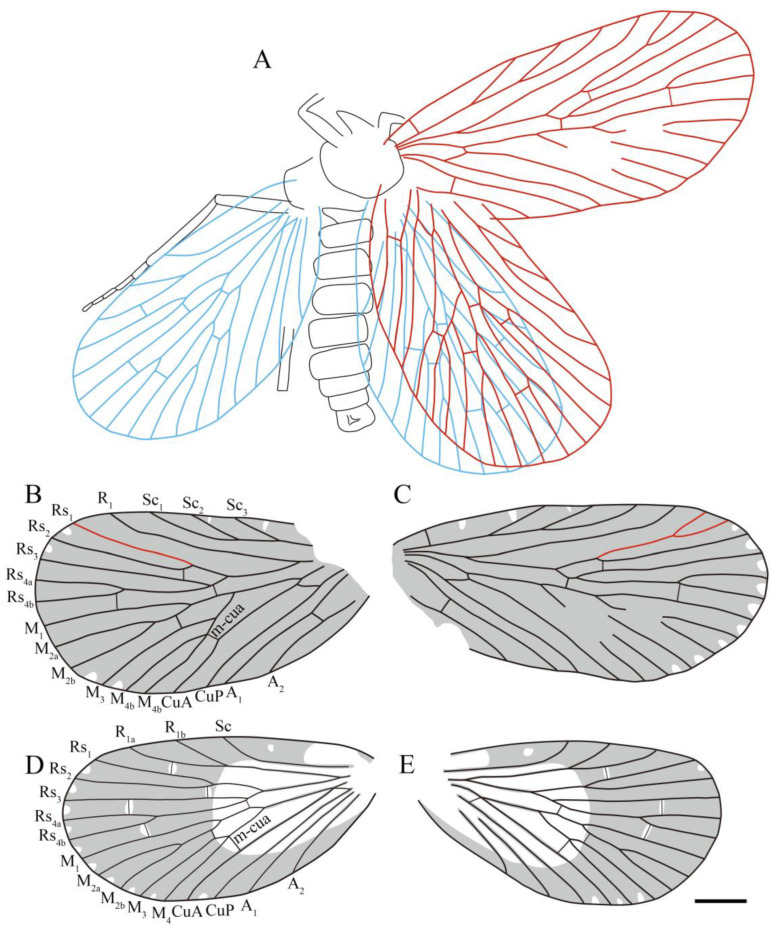
Line drawings of *Sinoagetopanorpa lini* sp. nov., NIGP200916 (holotype). (**A**) Line drawing of general habitus; (**B**) Line drawing of left forewing; (**C**) Line drawing of right forewing; (**D**) Line drawing of left hind wing; (**E**) Line drawing of right hind wing. Scale bar represents 1 mm.

**Figure 8 insects-14-00096-f008:**
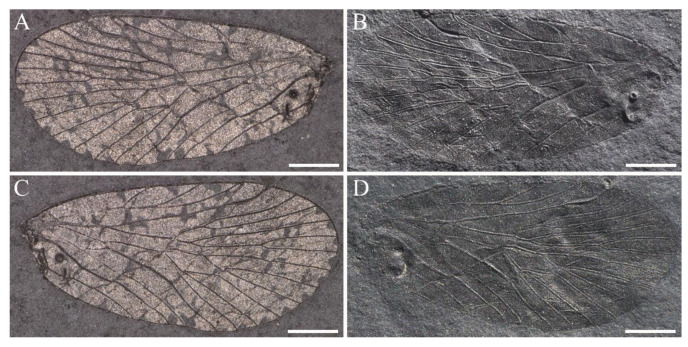
*Sinoagetopanorpa lini* sp. nov., NIGP200917 (paratype). (**A**,**B**) Photographs of part; (**C**,**D**) Photographs of counterpart; (**A**,**C**) were taken when specimens were immersed under 70% alcohol in vertical reflected light; (**B**,**D**) were taken in oblique reflected light. Scale bars represent 1 mm in (**A**–**D**).

**Figure 9 insects-14-00096-f009:**
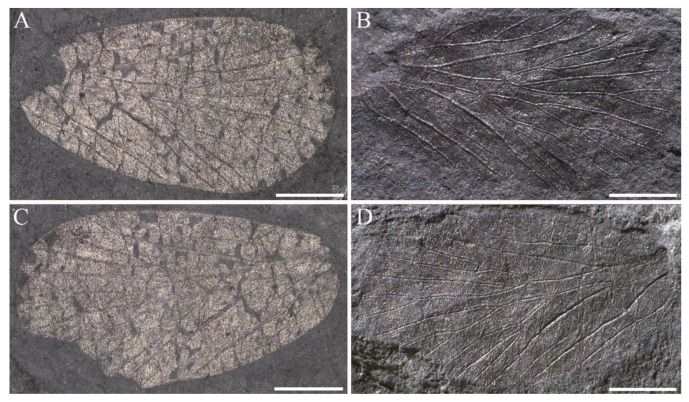
*Sinoagetopanorpa minuta* sp. nov., NIGP200920 (holotype). (**A**,**B**) Photographs of part; (**C**,**D**) Photographs of counterpart; (**A**,**C**) were taken when specimens were immersed under 70% alcohol in vertical reflected light; (**B**,**D**) were taken in oblique reflected light. Scale bars represent 1 mm in (**A**–**D**).

**Figure 10 insects-14-00096-f010:**
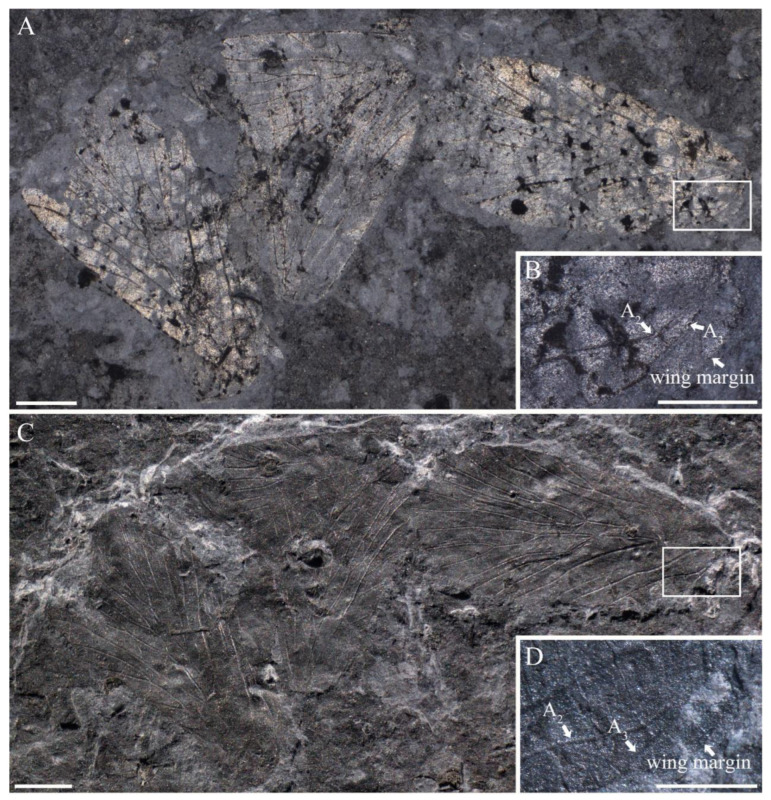
*Sinoagetopanorpa elegans* sp. nov., NIGP200921 (holotype). (**A**) Photograph was taken when specimen immersed under 70% alcohol in vertical reflected light; (**B**) Enlargement from (**A**); (**C**) Photograph was taken in oblique reflected light; (**D**) Enlargement from (**C**). Scale bars represent 1 mm in (**A**,**C**), 0.5 mm in (**B**,**D**).

**Figure 11 insects-14-00096-f011:**
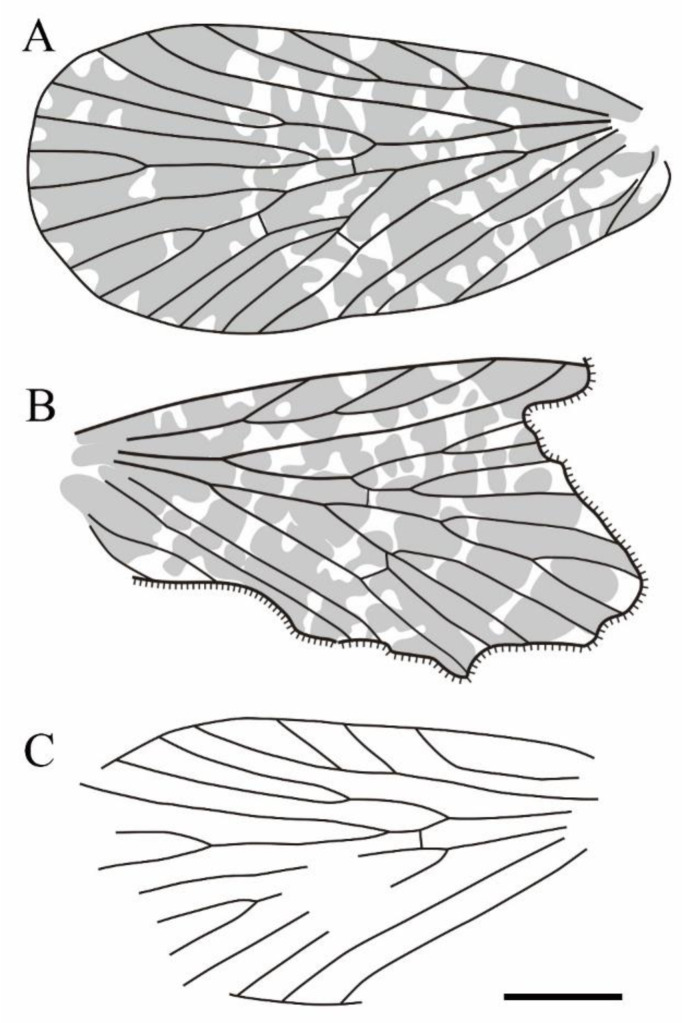
Line drawing of *Sinoagetopanorpa elegans* sp. nov., NIGP200921 (holotype). (**A**) Line drawing of one forewing; (**B**) Line drawing of the other forewing; (**C**) Line drawing of one hind wing, dark color not illustrated. Scale bar represents 1 mm.

**Figure 12 insects-14-00096-f012:**
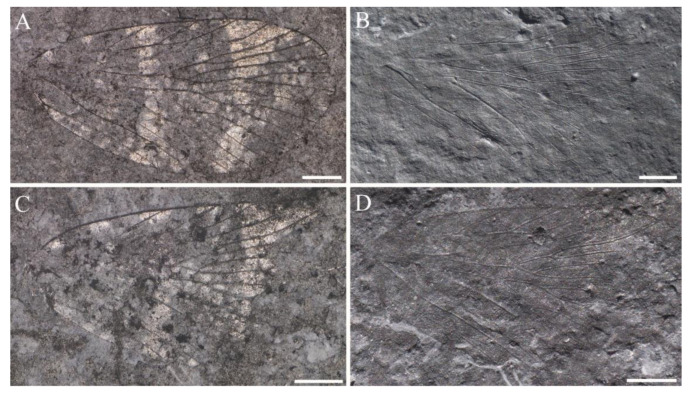
*Sinoagetopanorpa grimaldii* sp. nov. (**A**,**B**) Photographs of NIGP200922 (holotype); (**C**,**D**) Photographs of NIGP200923 (paratype); (**A**,**C**) were taken when specimens were immersed under 70% alcohol in vertical reflected light; (**B**,**D**) were taken in oblique reflected light. Scale bars represent 1 mm in (**A**–**D**).

**Figure 13 insects-14-00096-f013:**
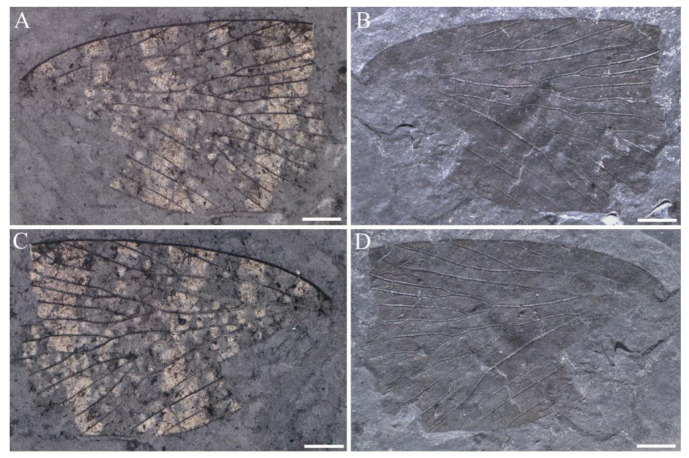
*Sinoagetopanorpa magna* sp. nov., NIGP200926 (holotype). (**A**,**B**) Photographs of part; (**C**,**D**) Photographs of counterpart; (**A**,**C**) were taken when specimens were immersed under 70% alcohol in vertical reflected light; (**B**,**D**) were taken in oblique reflected light. Scale bars represent 1 mm in (**A**–**D**).

**Figure 14 insects-14-00096-f014:**
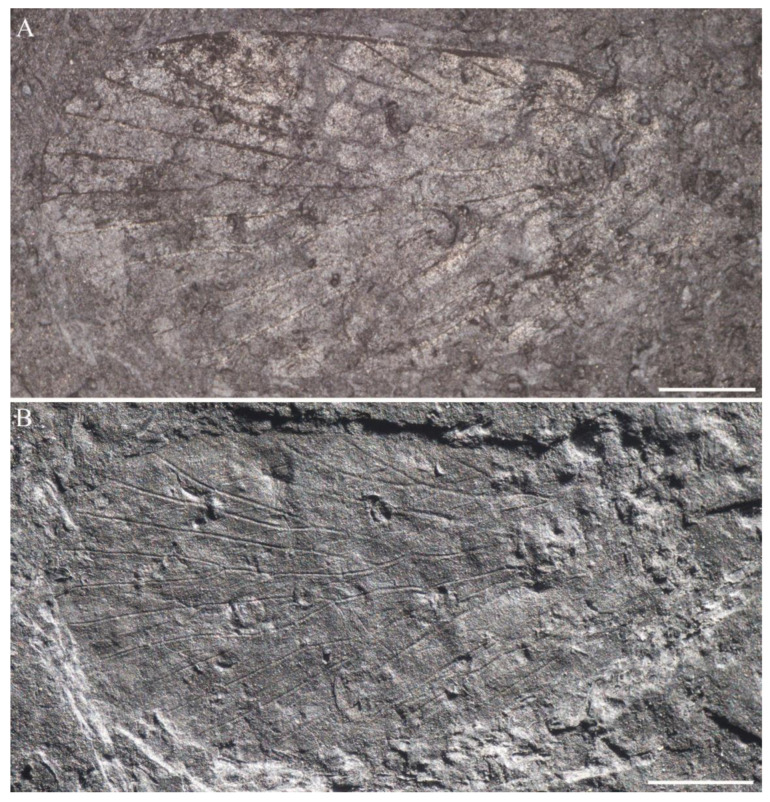
*Raragetopanorpa zhangi* sp. nov., NIGP200927 (holotype). (**A**) Photograph was taken when specimen was immersed under 70% alcohol in vertical reflected light; (**B**) Photograph was taken in oblique reflected light. Scale bars represent 1 mm in (**A**,**B**).

**Figure 15 insects-14-00096-f015:**
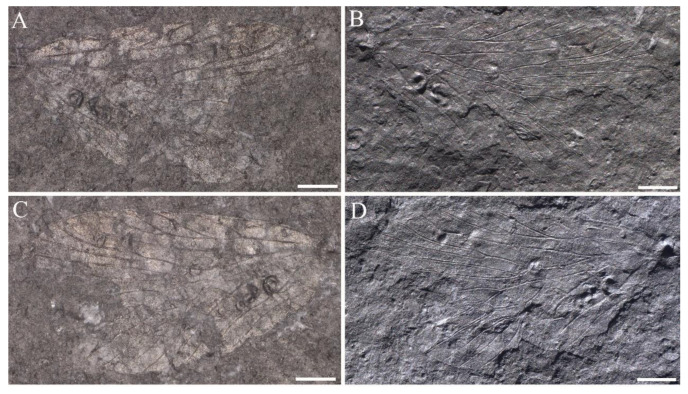
*Permoagetopanorpa yinpingensis* gen. et sp. nov., NIGP200928 (holotype). (**A**,**B**) Photographs of part; (**C**,**D**) Photographs of counterpart; (**A**,**C**) were taken when specimens were immersed under 70% alcohol in vertical reflected light; (**B**,**D**) were taken in oblique reflected light. Scale bars represent 1 mm in (**A**–**D**).

**Figure 16 insects-14-00096-f016:**
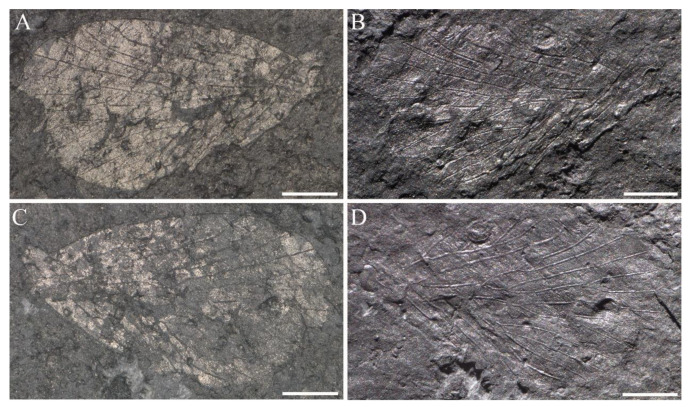
*Permoagetopanorpa incompleta* sp. nov., NIGP200929 (holotype). (**A**,**B**) Photographs of part; (**C**,**D**) Photographs of counterpart; (**A**,**C**) were taken when specimens were immersed under 70% alcohol in vertical reflected light; (**B**,**D**) were taken in oblique reflected light. Scale bars represent 1 mm in (**A**–**D**).

**Figure 17 insects-14-00096-f017:**
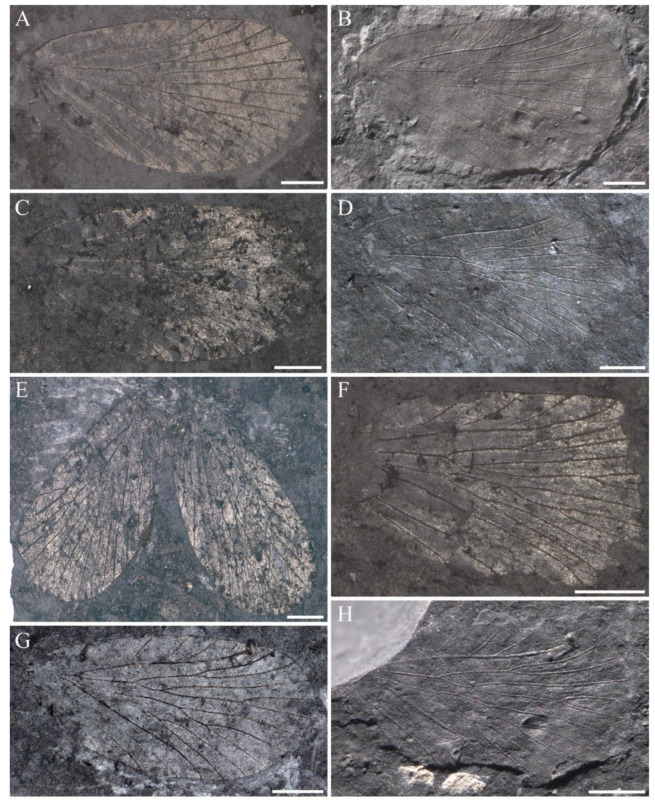
Hind wings of Sinoagetopanorpidae fam. nov. (**A**,**B**) Photographs of NIGP200930; (**C**,**D**) Photographs of NIGP200931; (**E**) Photograph of a pair of hind wings (NIGP200932); (**F**) Photograph of NIGP200933; (**G**) Photograph of NIGP200934 (mirror image); (**H**) Photograph of a NIGP200935; (**A**,**C**,**E**–**G**) were taken when specimens were immersed under 70% alcohol in vertical reflected light; (**B**,**D**,**H**) were taken in oblique reflected light. Scale bars represent 1 mm in (**A**–**H**).

**Figure 18 insects-14-00096-f018:**
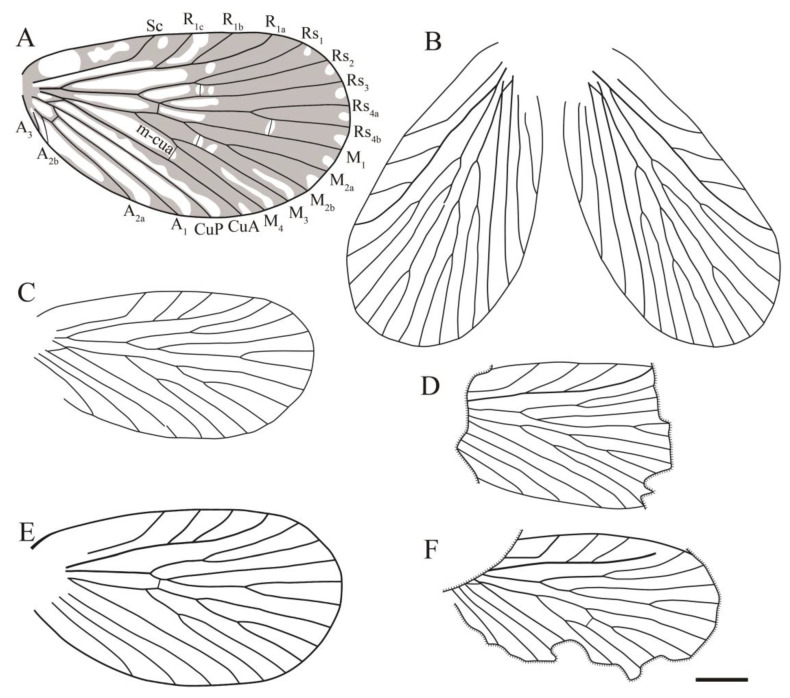
Line drawings of hind wings of Sinoagetopanorpidae fam. nov. (**A**) NIGP200930; (**B**) NIGP200932; (**C**) NIGP200934 (mirror image); (**D**) NIGP200933; (**E**) NIGP200931; (**F**) NIGP200935; (**B**–**F**) with dark color unillustrated. Scale bar represents 1 mm.

## Data Availability

All data generated during this study are included in this published article. All the specimens are housed in the Nanjing Institute of Geology and Palaeontology, Chinese Academy of Sciences, Nanjing, China.

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
