# Peer review of "Sinoagetopanorpidae fam. nov., a New Family of Scorpionflies (Insecta, Mecoptera) from the Guadalupian of South China"

_insects, 2023, doi:10.3390/insects14010096_

Round 1

Reviewer 1 Report

The paper is important to specialists of the Mecoptera. As this group is in the centre of interest of palaeoentomologists, it will attract also other specialists of the Mecopterida. The paper is carefully and beautifully illustrated; however, the descriptions should be rearranged, as listed below. The language is not good; I corrected only some mistakes.

A detailed comment follows.

Line 83: in the family diagnosis male genitalia are listed as diagnostic characters; in what genus and species they were found? Reference?

84: „a rounded triangle at apex of interface between each Rs and M branches” - an "interface" is probably used instead of "cell"?? As this seems to be one of the most characteristic key features of the family, its presence/absence in the species diagnoses should be supplemented (S. grimaldi and magna).

91: generic composition of a new family is missing.

102: name of taxon in italic

104, and further descriptions: just below the taxon name comes the diagnosis, then etymology; other sections come later. This arrangement is used in most taxonomic papers, as the diagnosis is the most important section.

182: these remarks should be included in the diagnosis (Differential diagnosis, recommended by the International Code of Zool. Nomenclature). This applies to all diagnoses.

397: as the species is dedicated to David Grimaldi, his name should be in the genitive in the species name, viz., grimaldii.

463: magna

474: Add: genus monotypic, and the next line is not necessary (redundant).

579: poor preservation

607: change to: Key to genera and species (as the key includes 3 genera)

608: at end of this line add „genus Permoagetopanorpa”

614: similarly at end of this line the genus Sinoagetopanorpa should be introduced. Such arrangement allows to discern the genera before species are separated further on.

659: shaped

676: distinguish

677: delete „an”

701: gone, not went

Finally, the zoobank numbers should be introduced for new taxa.

Signed: Ewa Krzemińska

Author Response

Reviewer 1

A detailed comment follows.

Line 83: in the family diagnosis male genitalia are listed as diagnostic characters; in what genus and species they were found? Reference?

The male genitalia is found in the holotype of the species Sinoagetopanorpa lini sp. nov. (Figures 6A-D, 7A)

84: „a rounded triangle at apex of interface between each Rs and M branches” - an "interface" is probably used instead of "cell"?? As this seems to be one of the most characteristic key features of the family, its presence/absence in the species diagnoses should be supplemented (S. grimaldi and magna).

It is better to use “interface’, for cell is not commonly used in Mecoptera. And we have supplemented to the diagnosis of S. grimaldi and S.magna.

91: generic composition of a new family is missing.

Added.

102: name of taxon in italic

Done.

104, and further descriptions: just below the taxon name comes the diagnosis, then etymology; other sections come later. This arrangement is used in most taxonomic papers, as the diagnosis is the most important section.

The order of those parts is based on the Insects style, so it is better to maintain it.

182: these remarks should be included in the diagnosis (Differential diagnosis, recommended by the International Code of Zool. Nomenclature). This applies to all diagnoses.

All of remarks concern with species comparison have deleted.

397: as the species is dedicated to David Grimaldi, his name should be in the genitive in the species name, viz., grimaldii.

Thanks, done.

463: magna

Done.

474: Add: genus monotypic, and the next line is not necessary (redundant).

Done.

579: poor preservation

Done.

607: change to: Key to genera and species (as the key includes 3 genera)

Done.

608: at end of this line add, enus Permoagetopanorpa”

Done.

614: similarly at end of this line the genus Sinoagetopanorpa should be introduced. Such arrangement allows to discern the genera before species are separated further on.

Thanks, the keys to genera have been added.

659: shaped

Done.

676: distinguish

Done.

677: delete „an”

Done.

701: gone, not went

Done.

Finally, the zoobank numbers should be introduced for new taxa.

Done.

The age of the Daohugou biota

Many authors used the Middle Jurassic for the age of the Daohugou fauna, including us. We know the age of the Daohugou beds in fact across the Middle-Late Jurassic boundary that confirmed by our recent CA-ID-TIMS dating. The age of Middle-Late Jurassic for the Daohugou fauna have been used in our many publications.

Reviewer 2 Report

In the manuscript, Lian et al. describe three genera and eleven species belonging to a new family of Mecoptera. The manuscript is well-prepared with many beautiful images. I recommend its acceptance in the Journal Insects after only a few small revisions.

Line 37.

Mecoptera are, one of, the most ancient holometabolous orders. Rephrase this sentence.

Line 80.

Add suborder and/or superfamily, if applicable.

Line 83.

This part needs to be more concise. Some phrases such as "small scorpionflies" obviously provides no diagnostic information, since many other families can also be small. If the family is currently established by a single genus, then the diagnosis should be identical for the new family and the type genus.

Line 91.

Add a "genera included" part to inform the readers how many genera in your family.

Line 102.

Italics for the Latin name.

Line 182.

The "remarks" part seems merely a repetition of the diagnosis, providing no new information. Why not rewrite this part?

Line 192.

What is a "triangle"? Is it a triangular spot? Or a hyaline, triangular spot?

Line 230.

Same problem as the preceding species. Why not combine "diagnosis" and "remarks"? Rewrite.

Line 251.

The correct plural form for thorax is thoraces. Since there are only one head, one thorax, and one abdomen in each insect, why do you use plural thoraces? Do you mean the thoracic segments?

Line 302.

Same as above.

Line 342.

Same as above.

Line 607.

The layout of the key is somehow in a jumble. The Latin names are not in italics.

Author Response

Reviewer 2

Line 37.

Mecoptera are, one of, the most ancient holometabolous orders. Rephrase this sentence.

done

Line 80.

Add suborder and/or superfamily, if applicable.

 It is a very constructive suggestion. Though some researchers supposed several divisions on suborder rank, most of the Permian families are erected based on isolated wings, and many authors believed that it is not plausible to erect a higher rank out of family.

Line 83.

This part needs to be more concise. Some phrases such as "small scorpionflies" obviously provides no diagnostic information, since many other families can also be small. If the family is currently established by a single genus, then the diagnosis should be identical for the new family and the type genus.

 Thanks, we have revised.

Line 91.

Add a "genera included" part to inform the readers how many genera in your family.

 Done.

Line 102.

Italics for the Latin name.

 Done.

Line 182.

The "remarks" part seems merely a repetition of the diagnosis, providing no new information. Why not rewrite this part?

 It is very good suggestion, and those remarks look somewhat redundant, we decided to delete those species comparison in “remarks”.

Line 192.

What is a "triangle"? Is it a triangular spot? Or a hyaline, triangular spot?

They are hyaline triangular spot, and all of concerned context has changed.

Line 230.

Same problem as the preceding species. Why not combine "diagnosis" and "remarks"? Rewrite.

 Done.

Line 251.

The correct plural form for thorax is thoraces. Since there are only one head, one thorax, and one abdomen in each insect, why do you use plural thoraces? Do you mean the thoracic segments?

We mean the whole thorax, and thoraces have changed to thorax.

Line 302.

Same as above.

  Done.

Line 342.

Same as above.

  Done.

Line 607.

The layout of the key is somehow in a jumble. The Latin names are not in italics.

Thanks, we have revised them.

Reviewer 3 Report

I have reviewed the manuscript Insects-2136696: Sinoagetopanorpidae fam. nov., a new family of scorpionflies (Insecta, Mecoptera) from the Guadalupian of South China” and submitted my review, revisions, and comments on January 5, 2023.

In this manuscript, they described and illustrated three genera (two new genera) and eleven species (ten new species) belonging to a new family Sinoagetopanorpidae fam. nov. from the upper Guadalupian Yinping Formation of Anhui Province, China. Their new discovery indicates a high diversity of mecopterans in the Permian of China, and Signoagetopanorpidae might have evolved independently on the Yangtzi Platform.

The co-authors have done a good job collecting many fossil specimens from this locality, analyzing morphological characters, setting up a key to species of Sinoagetopanorpidae fam nov., and comparing this new family with members of the subfamily Agetopanorpinae of Permochoristidae and members of the family Choristopsychidae. They have provided clear and detailed data and comparisons of the new specimens with two types of photos and line drawings. 

Here are some key revisions and suggestions:

·        The title states “South China”, but in the Abstract and Materials and Methods, they clearly state “Anhui Province, eastern China”. They need to be consistent and change the title to “Eastern China”.

·        Three words used in the list of Keywords need to be revised.

·        Change all “hindwing(s)” to “hind wing(s)”.

·        Change “Permoagetopanorpa incompleta gen. et sp. nov.” to “Permoagetopanorpa incompleta sp. nov.”

·        Make sure all names of genus and species are in Italic font.

·        Upgrade English writing.  

In the attached PDF version of the manuscript, I used the Open Comments and yellow highlights to indicate suggested revisions and improvements for this paper.

Author Response

Reviewer 3

Here are some key revisions and suggestions:

  • The title states “South China”, but in the Abstract and Materials and Methods, they clearly state “Anhui Province, eastern China”. They need to be consistent and change the title to “Eastern China”.

We have used South China for all.

  • Three words used in the list of Keywords need to be revised.

Done.

  • Change all “hindwing(s)” to “hind wing(s)”.

Done

  • Change “Permoagetopanorpa incompletagen. et sp. nov.” to “Permoagetopanorpa incompleta sp. nov.”

Done

  • Make sure all names of genus and species are in Italic font.

Done

  • Upgrade English writing.  

Thanks, we have revised as hard as we can.